# The Growth Mechanism of Boron-Doped Diamond in Relation to the Carbon-to-Hydrogen Ratio Using the Hot-Filament Chemical Vapor Deposition Method

**DOI:** 10.3390/mi16070742

**Published:** 2025-06-25

**Authors:** Taekyeong Lee, Miyoung You, Seohan Kim, Pungkeun Song

**Affiliations:** 1Department of Materials Science and Engineering, Pusan National University, Busan 46241, Republic of Korea; ltk9512@pusan.ac.kr; 2The Institute of Materials Technology, Pusan National University, Busan 46241, Republic of Korea; 3Department of Materials Science and Engineering, Ångström Laboratory, Uppsala University, P.O. Box 35, SE-751 03 Uppsala, Sweden

**Keywords:** hot-filament chemical vapor deposition (HF-CVD), boron-doped diamond (BDD), carbon-to-hydrogen ratio (C/H ratio)

## Abstract

This study synthesized boron-doped diamond (BDD) thin films using hot-filament chemical vapor deposition at different carbon-to-hydrogen (C/H) ratios in the range of 0.3–0.9%. The C/H ratio influence, a key parameter controlling the balance between diamond growth and hydrogen-assisted etching, was systematically investigated while maintaining other deposition parameters constant. Microstructural and electrochemical analysis revealed that increasing the C/H ratio from 0.3% to 0.7% led to a reduction in sp2-bonded carbon and enhanced the crystallinity of the diamond films. The improved conductivity under these conditions can be attributed to effective substitutional boron doping. Notably, the film deposited at a C/H ratio of 0.7% exhibited the highest electrical conductivity and the widest electrochemical potential window (2.88 V), thereby indicating excellent electrochemical stability. By contrast, at a C/H ratio of 0.9%, the excessively supplied carbon degraded the film quality and electrical and electrochemical performance, which was owing to the increased formation of sp^2^ carbon. In addition, this led to an elevated background current and a narrowed potential window. These results reveal that precise control of the C/H ratio is critical for optimizing the BDD electrode performance. Therefore, a C/H ratio of 0.7% provides the most favorable conditions for applications in advanced oxidation processes.

## 1. Introduction

Diamond is a crystal of carbon characterized by a tetrahedral sp^3^-bonded structure, resulting in an extremely rigid three-dimensional lattice [1,2,3]. Diamonds exhibit a wide band gap (~5.5 eV) because of their strong covalent bonding and symmetrical crystal arrangement, thereby providing excellent insulating properties and transparency across a broad spectral range [2,4,5,6,7]. Moreover, the strong covalent bonds between carbon atoms impart exceptional chemical inertness and outstanding physical durability, which enable diamonds to withstand harsh environmental conditions, such as extreme temperature fluctuations and exposure to reactive chemicals [8,9,10]. According to these exceptional physical and chemical properties, diamonds have been developed in various structural forms to meet various application needs, such as polycrystalline diamond, diamond-like carbon, and single-crystal diamond, which have been tailored on the basis of specific device requirements and targeted functionalities [11,12,13]. Each of these forms offers distinct characteristics suitable for specific purposes such as wear-resistant coatings, electronic substrates, and electrochemical electrodes, thereby expanding the practical applications of diamond-based materials across multiple technological fields [1,7,10,14].

Among them, polycrystalline diamonds have attracted substantial attention and have been extensively investigated. Intrinsic diamonds exhibit extremely high electrical resistivity (~10^16^ Ω·cm), which limits their application in electronic materials [1,15]. However, polycrystalline diamonds can exhibit electrical conductivity through substitutional impurity doping, such that nitrogen and phosphorus can generate n-type diamond and boron can generate p-type diamond [2,3,6,8,16]. In particular, boron has a relatively low activation energy (~0.37 eV) and can be effectively substituted at carbon sites within the diamond lattice, thereby enabling high doping concentrations [6]. Boron-doped diamonds (BDDs) have been extensively investigated and used in electrochemical applications because of their excellent chemical and physical durability and broad electrochemical potential window, indicating their ability to produce strong oxidants that can degrade non-degradable organics [1,2,3,6,14,17,18].

Several representative polycrystalline diamond deposition methods exist, including hot-filament chemical vapor deposition (HF-CVD), microwave plasma CVD, radio-frequency plasma CVD, and direct current plasma CVD [19,20,21,22]. HF-CVD is especially widely used because of its simple equipment configuration, low cost, and suitability for large-area deposition [19,22]. In the case of HF-CVD, the diamond film is deposited according to the following fundamental steps of diamond synthesis: (1) initiating the thermal decomposition of hydrogen gas, which is dissociated into atomic hydrogen, (2) CH_4_ decomposes and provides reactive carbon species, (3) the substrate is heated by radiative heating from the filament, (4) reactive carbon species adsorb onto the substrate surface; mainly near diamond seed, (5) non-diamond carbon (sp^2^) is preferentially etched, while diamond (sp^3^) is stabilized, promoting nucleation and growth [7,19,23]. Based on these process steps, diamonds get synthesized, with the synthesis parameters determining the details of the crystallinity, grain size, and crystal facet [4,14,24]. Synthesis parameters are deeply intertwined with each other and play a key role because they considerably affect the crystallinity, doping level, morphology, and overall quality of deposited diamond films [5,8,24,25,26]. In particular, for BDD electrodes, the presence of sp^2^-bonded carbon within the diamond lattice or on the surface can negatively affect electrochemical behavior by narrowing the potential window and increasing the background current [2,27].

Herein, we specifically focused on the effect of varying the carbon-to-hydrogen (C/H) ratio on the crystallinity of polycrystalline diamond films, influencing the quality of BDD. All other experimental parameters, including the deposition pressure, filament temperature, and substrate–filament distance, were carefully fixed to isolate the influence of the C/H ratio. As aforementioned, the diamond growth mechanism using HF-CVD is mainly governed by critical deposition parameters, such as the C/H ratio, deposition pressure, substrate and filament temperature, and substrate-to-filament distance [4,14,24]. Among these, the C/H ratio is specifically important because it directly influences the competitive processes of diamond growth and the hydrogen-induced etching of sp^2^-bonded carbon. Hydrogen (H_2_), introduced as a reactive gas, is responsible for the suppression of sp^2^-bonded carbon and its preferential etching and influences the grain growth of diamond films [24,25,28]. In addition, sp^2^ carbon preferentially forms at diamond grain boundaries; thus, grain size is a key factor affecting the diamond film quality and BDD electrode performance [16,29]. Therefore, a proper balance between carbon and hydrogen content is essential for achieving high-quality polycrystalline diamond films with few structural defects, enhanced crystallinity, and optimal doping efficiency. We systematically investigated the effect of the C/H ratio on diamond crystallinity and subsequently explored the correlation between these structural changes and variations in the electrochemical performance of the fabricated BDD electrodes. This study provides insights into the optimization of diamond deposition conditions for enhanced electrode properties suitable for electrochemical degradation.

## 2. Experimental

### 2.1. BDD Thin Film Deposition

BDD thin films were deposited on niobium and silicon substrates using the HF-CVD method with tantalum filaments. This study aimed to evaluate the performance of BDD as an electrochemical electrode for advanced oxidation processes. Niobium was selected as the substrate because of its high melting point, thermal and chemical stability, high electrical conductivity, and mechanical properties, including strong resistance to oxidation and acid corrosion. Silicon was also selected as a substrate material because of its ease of handling, low ductility, and small thermal expansion coefficient. An alumina substrate was also used for Hall effect measurements to eliminate the influence of the substrate on the Hall effect due to its electrically insulating properties.

Tantalum filaments measuring 0.7 mm in diameter and 230 mm in length were used in the HF-CVD process, with 11 filament lines installed. The distance between the filament and substrate was fixed at 9 mm. All samples were deposited for 10 h, and the deposition pressure was maintained at 30 Torr. During deposition, the filament temperature was maintained at 2400 °C, and the substrate temperature was approximately 950 °C. The filament temperature was measured using a 2-color pyrometer (E1RH-F1-1-0, FLUKE, Everett, WA, USA), and the substrate temperature was monitored using a Type R thermocouple(Daejeon, South Korea). For diamond growth, methane gas (CH_4_, 99.95%) was used as the carbon source, hydrogen (H_2_, 99.999%) was used as the reactive gas, and trimethyl boron (TMB, B(CH_3_)_3_, 1000 ppm in H_2_) was used as the boron source. In this study, we focused on the quality of BDD as a function of the C/H ratio. TMB gas mixture contains a carbon source (CH_3_); thus, the carbon contributions from both CH_4_ and TMB were considered when calculating the C/H ratio. The boron-to-carbon (B/C) ratio was kept constant across all samples to isolate the effect of the C/H ratio. The deposition conditions are detailed in Table 1.

### 2.2. BDD Thin Film Characterization

Surface morphology and cross-sectional images of the BDD thin films were obtained using field-emission scanning electron microscopy (FE-SEM, Mira 3, Tescan, Brno, Czech Republic) operated at an accelerating voltage of 10 kV. The microstructure of the thin films was analyzed via X-ray diffraction (XRD, Ultima IV, Rigaku, Tokyo, Japan) employing Cu Kα radiation (λ = 0.154 nm). XRD measurements were performed over a 2θ range of 20–80° with a scan step of 0.02° and a scan speed of 2°/min. The operating voltage and current were set to 40 kV and 40 mA, respectively. Raman spectroscopy (HEDA, WEVE) was performed using a 532 nm excitation laser to confirm boron doping and the presence of sp^2^-bonded carbon in the films. The chemical states of carbon and boron were investigated via X-ray photoelectron spectroscopy (XPS, ESCALAB 250Xi, Thermo Fisher Scientific, Waltham, MA, USA). The binding energy was calibrated based on the C-C contribution, which was derived from the C 1s adventitious carbon signal at 284.8 eV, and the obtained data were handled using CasaXPS software(Version 2.3.19PR1.0) [30]. The electrical properties of the BDD films were evaluated using a Hall effect measurement system (HMS-3000, ECOPIA, Anyang-si, South Korea). The electrochemical potential window of the thin films was measured via cyclic voltammetry (CV) using a potentiostat (VersaSTAT 3, Princeton Applied Research, AMETEK, Oak Ridge, TN, USA). A 0.1 M H_2_SO_4_ solution was used as the electrolyte, and measurements were performed at a scan rate of 1 V/s in the potential range of −3.0–3.0 V. A platinum wire and saturated calomel electrode were used as the counter and reference electrodes, respectively.

## 3. Results and Discussion

Figure 1a–d show the surface morphology of the deposited BDD thin films with different C/H ratios, as observed via FE-SEM. All samples exhibited continuously formed films, and in the samples with a C/H ratio of up to 0.7%, distinct polyhedral-shaped grains were formed. The grain size of the diamond films gradually increased as the C/H ratio increased. However, the film with a C/H ratio of 0.9% exhibited a cauliflower-like surface morphology. The presence of reactive carbon species at an appropriate level ensures a balance between nucleation and nuclei growth, resulting in a well-developed diamond film. By contrast, an excessive supply of reactive carbon species beyond a threshold destroys the balance between nucleation and the growth of nuclei, thereby forming sp^2^-bonded carbon. In addition, the oversupply induces dense nucleation and insufficient diamond growth time, thereby forming ultrafine sub grains. As shown in Figure 1, the deposited film with a C/H ratio of 0.9% exhibits very fine sub grains grown on large grains, indicating that the deposition occurred beyond the content threshold of reactive carbon species. Moreover, the carbon species content influenced the deposition rate. The results are presented in Figure 1e–h. When the C/H ratio was increased from 0.3% to 0.7%, the film thickness increased from 720 to 1270 nm, indicating an enhanced deposition rate owing to the increased carbon supply. However, at a C/H ratio of 0.9%, the film thickness decreased to 910 nm. As aforementioned, this behavior is attributed to excessive nucleation and the formation of sp^2^-bonded carbon, which hinders grain growth. The nuclei growth with high-density nucleation has severe competition with near grains, most of the nuclei fail to grow sufficiently, and only a few grow vertically from the substrate; the growth of the remaining grains is suppressed [31]. Consequently, fine grain structures resembling a cauliflower were formed on the surface, and the overall grain growth rate and deposition rate tended to decrease. This suggests that crystallinity was maintained during the initial growth stages of the film; however, as the growth progressed under an excessive carbon supply, the growth of grains, except those oriented vertically from the substrate, was gradually inhibited and sp^2^-bonded carbon was formed on the surface.

To investigate the microstructure of the BDD thin films in relation to their C/H ratio, XRD analysis was performed. The results are presented in Figure 2, and the crystallite sizes calculated using the Scherrer equation are listed in Table 2 [32]. The crystalline phase of BDD was confirmed by a prominent diffraction peak corresponding to the diamond (111) plane at 43.9°. This peak was observed in all samples, indicating that the continuous growth of diamond films occurred regardless of the C/H ratio. As the C/H ratio increased, the intensity of the diamond (111) peak gradually increased. With this preferred orientation of BDD, the crystallite size and full width at half maximum (FWHM) were calculated. No significant difference in crystallite size and FWHM was observed in relation to the C/H ratio. In contrast, at the highest C/H ratio (0.9%), the crystallite size significantly decreased. This trend is consistent with the surface morphology observations in Figure 1, where the grain size decreases and a cauliflower-like structure appears at a C/H ratio of 0.9%. In the case of BDD with a C/H ratio of 0.5%, the grain size observed in the SEM image was approximately 1 µm, whereas the crystallite size calculated from the XRD data was 24.33 nm, showing a considerable difference. There are two primary reasons for this discrepancy. First, the crystallite size calculated using the Scherrer equation reflects the coherent diffracting domain associated with a specific crystallographic plane, rather than the entire grain [33]. Second, XRD probes the interior of the film, whereas the morphological information provided by SEM is limited to the surface following complete film formation. These differences in measurement principles and perspectives account for the discrepancy between the two results.

In the Raman spectra (Figure 3), the sample deposited at a C/H ratio of 0.3% exhibited a distinct diamond characteristic peak at 1332 cm^−1^ [34,35,36], whereas weak bands associated with boron-induced scattering were observed at approximately 500 and 1200 cm^−1^ [5,36,37,38]. These results indicate that the sufficient substitutional incorporation of boron is not achieved under these conditions. As aforementioned, only the C/H ratio was used as a variable to investigate the microstructure and electrical and chemical properties of the BDD thin films. Notably, the flow rate of TMB, a gas mixture containing carbon and hydrogen, was changed to maintain the B/C flow ratio. Alternatively, even with a constant B/C ratio, the absolute amount of boron may be insufficient at lower C/H ratios, resulting in inadequate doping, which can be the reason for the weak doping-related features observed in Figure 3a. With an increasing boron concentration (Figure 3b–d), the characteristic diamond peak at 1332 cm^−1^ gradually decreased in intensity and broadened. This behavior is attributed to the enhanced lattice distortion and electron–phonon coupling induced by substitutional boron atoms. Furthermore, free carrier absorption considerably reduced the Raman scattering efficiency, resulting in additional attenuation of the diamond peak [37,39]. However, in the sample with a C/H ratio of 0.9%, a Raman band in the range of 1500–1550 cm^−1^ corresponding to sp^2^-bonded carbon was observed [34,35,40,41]. It is also notable that a minor band near 1550 cm^−1^ appeared in the sample with a C/H ratio of 0.3%, although its intensity was significantly lower than that observed at 0.9%, indicating the presence of a small amount of sp^2^-bonded carbon. XPS analysis was performed to thoroughly analyze the sp^2^ and sp^3^ bonding in the diamond films.

The XPS core-level deconvoluted spectra of BDD in relation to the C/H ratio for C 1s and B 1s are shown in Figure 4 and the corresponding quantitative analysis is summarized in Table 3. The C 1s core level was deconvoluted into sp^3^- and sp^2^-bonded carbon components [42,43]. In general, at diamond grain boundaries, carbon exists as a mixture of sp^2^ and sp^3^ bonds [16]. As the C/H ratio increased from 0.3% to 0.7% (Figure 4a–c), the amount of sp^2^-bonded carbon gradually decreased. The decrease in sp^2^-bonded carbon content within the C/H ratio range of 0.3–0.7% is attributed to an increase in the grain size accompanied by diamond growth, as shown in Figure 1a–c. In contrast, at a C/H ratio of 0.9%, the sp^2^-bonded carbon content significantly increased. This result supports the interpretation of Figure 1d, where the cauliflower-like morphology is attributed to the enhanced formation of sp^2^ carbon due to an oversupply of carbon species. In addition, this trend is consistent with the Raman spectroscopy results shown in Figure 3, particularly at 0.5% and 0.7% C/H ratios, where the stable presence of sp^3^-bonded carbon confirms the growth of highly crystalline diamond films. The B 1s peak was deconvoluted into B-B, B-C, B-H, and B-O bonds [44,45,46,47]. When boron was doped into the diamond system, it was predominantly incorporated via substitutional sites, forming B-C bonds with the carbon atoms in the lattice. As shown in Figure 4e–g, in samples with a moderate boron content, B-C bonding increased, indicating effective substitutional doping. In contrast, at a 0.9% C/H ratio (Figure 4h), B-C bonding decreased, because the decrease in the sp^3^-bonded carbon content reduced the number of lattice sites available for boron substitution. Boron atoms that were not substitutionally incorporated into sp^3^-bonded lattice sites may have formed unstable bonds with sp^2^-bonded carbon, leading to the formation of dangling bonds. These boron atoms may interact with each other, bond with abundant hydrogen species, or react with other boron atoms, potentially resulting in structural or chemical defects in the film. Consequently, increased B-H, B-O, and B-B bonding was observed.

Figure 5 and Table 4 presents the electrical properties of the deposited BDD thin films with different C/H ratios, as analyzed based on Hall effect measurements. According to the Hall measurement results, resistivity initially gradually decreased as the C/H ratio increased from 0.3% to 0.7% and then increased at a C/H ratio of 0.9%. The carrier concentration increased as the C/H ratio increased from 0.3% to 0.7%, which is consistent with the trend of B-C bonding observed in Figure 4e–g. The mobility exhibited a general decreasing trend with an increasing C/H ratio. Considering that all samples exhibited high carrier concentrations on the order of 10^20^ cm^−3^, ionized impurity scattering due to heavy boron doping is considered the primary factor contributing to reduced mobility [48].

In general, resistivity is inversely proportional to both the carrier concentration and mobility. However, in this study, although mobility decreased as the C/H ratio increased from 0.3% to 0.7%, resistivity decreased. This result is attributed to the dominant influence of high carrier concentrations (up to 10^20^ cm^−3^). In contrast, at a 0.9% C/H ratio, the reduced carrier concentration and grain size likely interfered with charge transport, resulting in improved resistivity [49]. Overall, the BDD film deposited at a C/H ratio of 0.7% exhibited the most superior electrical properties.

Figure 6 depicts the results of the CV analysis of the electrochemical properties of the BDD films deposited at various C/H ratios. Diamond possesses a chemically inert surface, preventing the adsorption of reactants. Thus, diamonds act as an inert electrode in electrochemical reactions, participating only in electron transfer without undergoing any chemical transformation. Because of these characteristics, diamond exhibits a wide and stable electrochemical potential window, referring to the voltage range in which no redox reactions occur, and serves as a key indicator of electrochemical stability. Herein, the potential window of each sample was analyzed from the CV curves, and horizontal arrows were added to indicate the flat current regions for a clearer visual comparison. According to the CV analysis, the BDD thin films deposited at C/H ratios of 0.5% and 0.7% exhibited the widest and stable potential windows, indicating superior electrochemical stability. By contrast, although the potential window values showed no significant difference, C/H ratios of 0.3% and 0.9% samples exhibited higher background currents and unstable current behavior within the range of −1.0 to 1.0 V. This slightly affected the effective potential window and indicates a reduction in electrochemical stability. These differences are strongly correlated with the quality of the deposited films and can be attributed to increased surface reactivity or the presence of structural and chemical impurities. At a C/H ratio of 0.3%, the limited carbon supply limited the growth of diamond grains, leading to grain sizes smaller than those observed at 0.5% and 0.7%. At a C/H ratio of 0.9%, an increase in the sp^2^-bonded carbon content was observed (Figure 4d). The presence of grain boundaries and sp^2^-bonded carbon, which contain delocalized π-electrons, facilitates non-Faradaic charge accumulation on the electrode surface, thereby increasing the capacitive current and promoting undesirable redox reactions, thereby reducing reduced electrochemical stability [3,6,50,51]. These factors contributed to the electrochemical behavior observed in Figure 6. Therefore, a C/H ratio of 0.7%, which exhibited a wide and stable anodic potential window of 2.88 V vs. SCE with a low background current, demonstrated excellent electrochemical stability. Such electrochemical characteristics enable the stable formation of hydroxyl radicals (•OH), which are strong oxidants with an oxidation potential of approximately +2.8 V, while it has a low background current, ensuring electrode stability [52,53]. Accordingly, a C/H ratio of 0.7% is considered the optimal condition for achieving the superior electrochemical performance and stability of BDD as an electrode material.

## 4. Conclusions

Herein, BDD films were synthesized through HF-CVD at various C/H ratios to investigate the effects of the carbon content on the structural, electrical, and electrochemical properties of BDD. The results revealed that the C/H ratio plays a key role in determining the film morphology, crystallinity, and doping characteristics. At a low C/H ratio of 0.3%, an insufficient carbon supply limited diamond grain growth, resulting in small grains, reduced film thickness, and weak boron incorporation, which led to inferior electrical properties. As the C/H ratio increased, grain growth was promoted and effective substitutional boron doping was achieved, thereby improving electrical conductivity and widening the electrochemical potential window. However, at an excessively high C/H ratio of 0.9%, an oversupply of reactive carbon species caused a breakdown in the balance between nucleation and grain growth, forming a cauliflower-like morphology with fine sub grains and increased sp^2^-bonded carbon content. This structural degradation hindered carrier transport and promoted surface reactivity, thereby narrowing the electrochemical potential window and reducing electrical and electrochemical performance. Among the investigated conditions, the BDD film grown at a C/H ratio of 0.7% exhibited the most favorable combination of high crystallinity, effective boron doping, low resistivity, and a wide potential window of 2.88 V. These results reveal that BDD has a high potential for applications requiring electrochemically stable electrode materials, such as advanced electrochemical systems and other electrochemical technologies.

## Figures and Tables

**Figure 1 micromachines-16-00742-f001:**
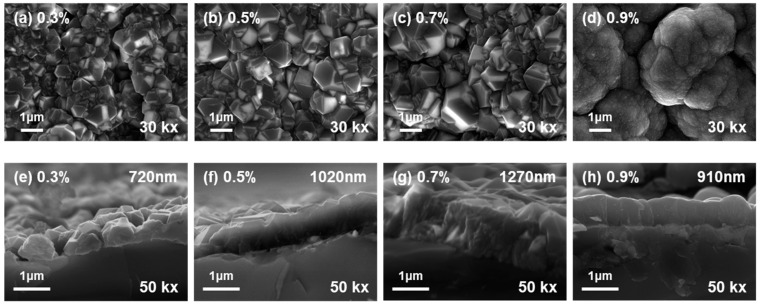
Surface (**a**–**d**) and cross-sectional images (**e**–**h**) of BDD thin films deposited in relation to the C/H ratio: (**a**,**e**) 0.3%, (**b**,**f**) 0.5%, (**c**,**g**) 0.7%, and (**d**,**h**) 0.9%.

**Figure 2 micromachines-16-00742-f002:**
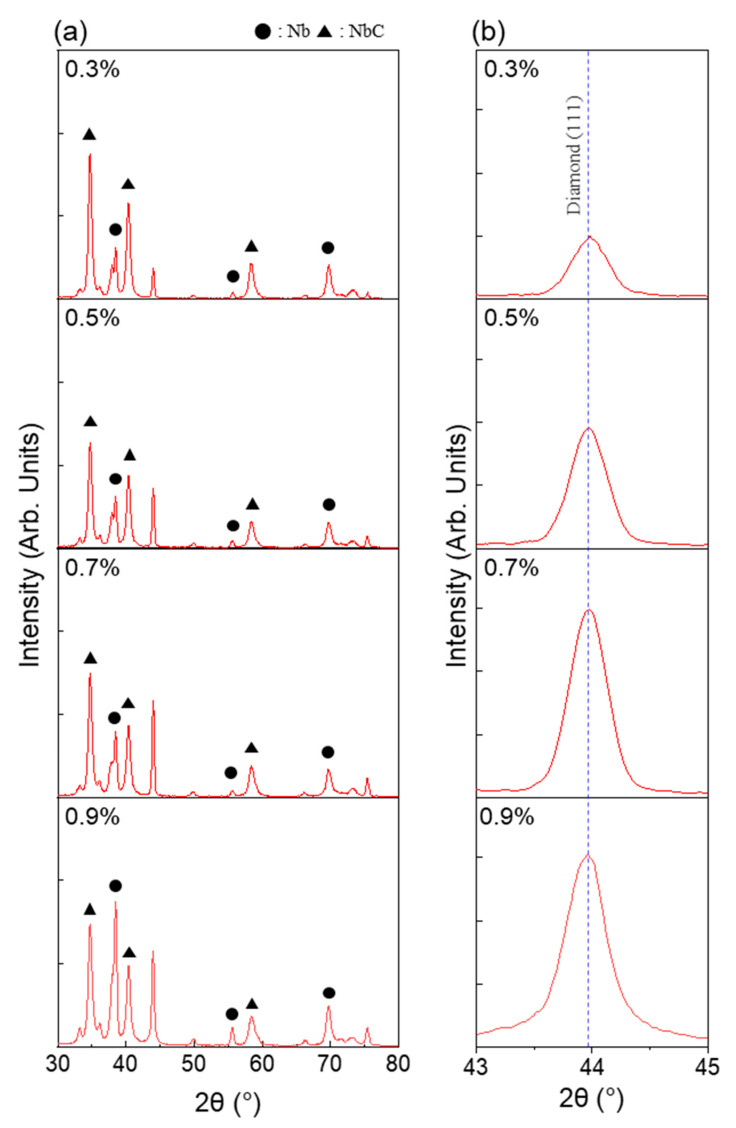
(**a**) XRD patterns of BDD thin films deposited with various C/H ratios, and (**b**) magnified view of diamond (111) peak (blue dashed line) at 43.9°.

**Figure 3 micromachines-16-00742-f003:**
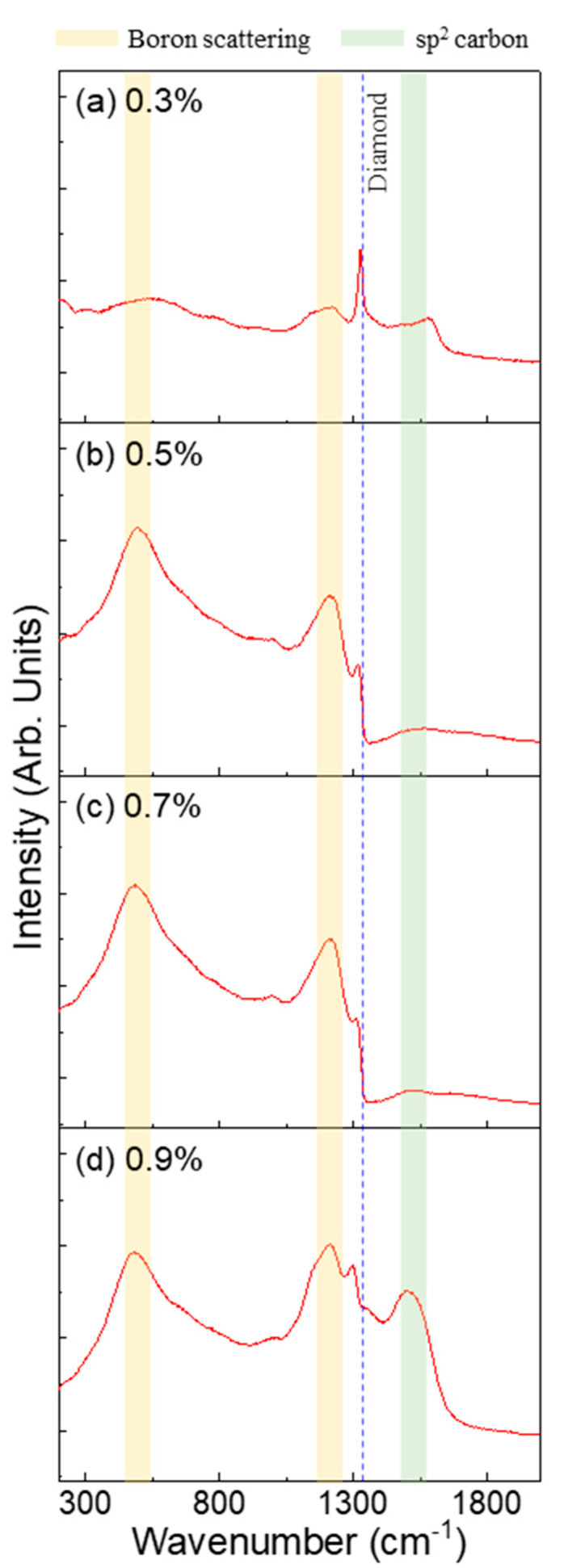
Raman spectra of BDD thin films deposited at different C/H ratios (0.3%, 0.5%, 0.7%, and 0.9%, top to bottom). Boron scattering and sp^2^ carbon regions are indicated in yellow and green, respectively.

**Figure 4 micromachines-16-00742-f004:**
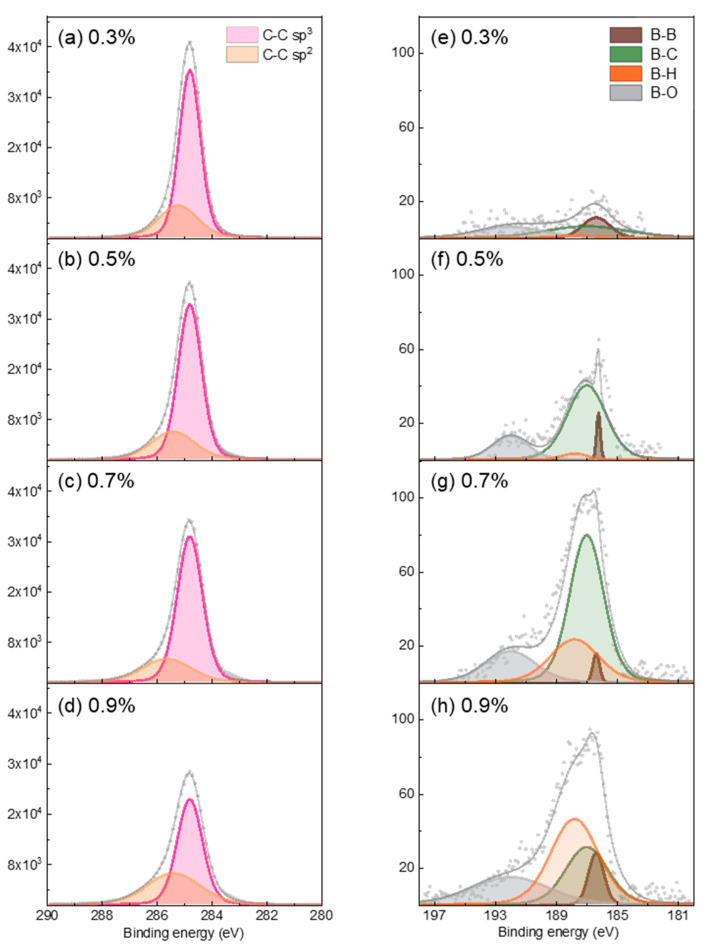
XPS core level spectra of BDD thin films in relation to the C/H ratio for (**a**–**d**) C 1s and (**e**–**h**) B 1s.

**Figure 5 micromachines-16-00742-f005:**
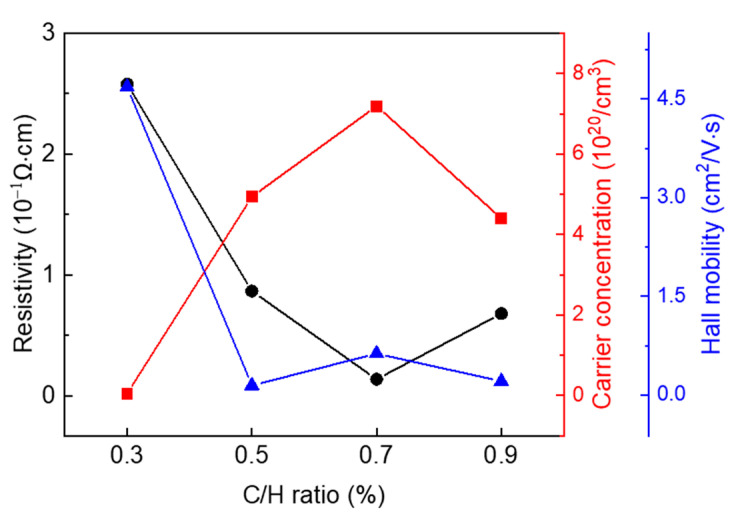
Electrical properties of BDD thin films deposited at various C/H ratios of 0.3%, 0.5%, 0.7%, and 0.9%.

**Figure 6 micromachines-16-00742-f006:**
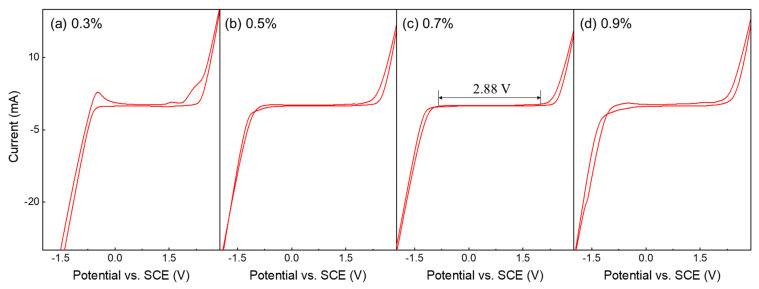
Cyclic voltammetry curves of BDD thin films deposited at various C/H ratios: 0.3%, 0.5%, 0.7%, and 0.9% (from left to right).

**Table 1 micromachines-16-00742-t001:** Experimental deposition conditions.

Parameter	Value
Filament	Tantalum
Substrate	Niobium, Silicon, Alumina
Carbon source	Methane (CH_4_)
Boron source	Trimethyl Boron (B(CH_3_)_3_)
Process pressure	30 Torr
Working distance	9 mm
Filament temp.	2400 °C
Substrate temp.	950 °C
H_2_ flow rate	450 sccm
CH_4_ flow rate	3, 5, 7, 9 sccm
TMB flow rate	3.5, 5.5, 8, 10 sccm
C/H ratio	0.3, 0.5, 0.7, 0.9%
B/C ratio	1100 ppm
Deposition time	10 h

**Table 2 micromachines-16-00742-t002:** FWHM, diamond (111) peak intensity, and diamond crystallite size of BDD films at various C/H ratios, calculated from the XRD data.

C/H Ratio	0.3%	0.5%	0.7%	0.9%
FWHM	0.36	0.37	0.37	0.42
Intensity	382.00	725.67	1161.00	1161.67
Grain size (nm)	24.89	24.33	24.01	21.20

**Table 3 micromachines-16-00742-t003:** Quantitative XPS analysis of BDD films at various C/H ratios using B 1s and C 1s spectra.

C/H Ratio	0.3%	0.5%	0.7%	0.9%
C-C sp^3^ (%)	73.76	74.51	75.40	60.57
C-C sp^2^ (%)	26.00	25.08	23.71	38.38
B-B (%)	0.06	0.02	0.02	0.09
B-C (%)	0.10	0.29	0.50	0.27
B-H (%)	0.01	0.02	0.16	0.45
B-O (%)	0.07	0.09	0.21	0.24

**Table 4 micromachines-16-00742-t004:** Electrical properties of BDD films at various C/H ratios, including resistivity, carrier concentration, and Hall mobility.

C/H Ratio	0.3%	0.5%	0.7%	0.9%
Resistivity (10^−1^ Ωcm)	2.58	0.87	0.14	0.68
Carrier Concentration (10^20^/cm^3^)	0.05	4.94	7.19	4.40
Hall mobility (cm^2^/Vs)	4.68	0.15	0.63	0.21

## Data Availability

The original contributions presented in this study are included in the article. Further inquiries can be directed to the corresponding authors.

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
