# Peer review of "The Growth Mechanism of Boron-Doped Diamond in Relation to the Carbon-to-Hydrogen Ratio Using the Hot-Filament Chemical Vapor Deposition Method"

_micromachines, 2025, doi:10.3390/mi16070742_

Round 1

Reviewer 1 Report

Comments and Suggestions for Authors

Overall, this work on regulating C/H in HF-CVD to fabricate BDD film electrode is a valuable study that may arouse the interest of relevant readers. The authors systematically evaluated the relationship between properties and material microstructure. The data are highly relevant to the explanations, and the explanations are clear and of a decent standard. However, there are still minor improvements to be made, such as the following:
1. Need to explain the peak close to sp2 band (1550 cm-1) in the Raman spectrum of 0.3% C/H sample.
2. The horizontal axis of the CV profile should indicate the standard potential to identify its position relative to the hydrogen reduction potential (H2/H+).
3. In addition, the author should indicate what reference potential +2.8V of forming .OH free radical is based on? If based on the reported CV profile, wouldn't the .OH free radical that can be generated occur in all C/H samples? (Every sample shows strong oxidation reaction beyond around 1.5 V, which is smaller than the 2.8 V mentioned in the manuscript, so every sample can generate .OH free radical at +2.8 V vs. the reported CV profile reference potential).

The oxidation position where .OH free radical occurs vs. the H2/H+ potential must be clearly defined. It is completely wrong to use the cell voltage of 2.88 to explain that it includes 2.8V. "Voltage" (Difference of potential) and "potential" (position) is totally a different thing. Anyway, the description here is not clear and is not supported by the shown data.

Author Response

Overall, this work on regulating C/H in HF-CVD to fabricate BDD film electrode is a valuable study that may arouse the interest of relevant readers. The authors systematically evaluated the relationship between properties and material microstructure. The data are highly relevant to the explanations, and the explanations are clear and of a decent standard. However, there are still minor improvements to be made, such as the following:

: Thank you very much for your kind and encouraging comments. We sincerely appreciate your positive evaluation of the significance and clarity of our work. The manuscript has been carefully revised in response to your suggestions. All changes have been highlighted in yellow background in the revised manuscript, and we have provided detailed, point-by-point responses to each of your comments below.

  1. Need to explain the peak close to sp2 band (1550 cm-1) in the Raman spectrum of 0.3% C/H sample.

Our response:

Thank you for your comment. In the original manuscript, we described the sp²-related Raman feature at approximately 1500 cm⁻¹. Upon review, this has been revised to a broader range of 1500–1550 cm⁻¹ to reflect its typical spectral shape, and supporting references have been added to substantiate this correction. Accordingly, the weak band near 1550 cm⁻¹ in the 0.3% C/H sample is attributed to sp²-bonded carbon. To make this clearer, we added the following sentence to clarify the presence of a weak sp²-related Raman band in the 0.3% C/H sample. This addition acknowledges the observed signal and provides a smooth transition to the subsequent XPS analysis, which further discusses the sp² and sp³ bonding states in the films.

New reference: (Line 218)

  1. Baron, C.; Ghodbane, S.; Deneuville, A.; Bustarret, E.; Ortega, L.; Jomard, F. Detection of CHx Bonds and Sp2 Phases in Polycrystalline and Ta-C:H Films from Raman Spectra Excited at 325 Nm. In Proceedings of the Diamond and Related Materials; March 2005; Vol. 14, pp. 949–953.
  2. Prawer, S.; Nugent, K.W.; Jamieson, D.N. RELATED TER|AL$ The Raman Spectrum of Amorphous Diamond; 1998;

Original: (Line 218)

However, in the sample with a C/H ratio of 0.9%, a Raman band at approximately 1500 cm−1 corresponding to sp2-bonded carbon was observed.

Revised: (Line 218)

However, in the sample with a C/H ratio of 0.9%, a Raman band in the range of 1500–1550 cm⁻¹ corresponding to sp²-bonded carbon was observed. It is also notable that a minor band near 1550 cm⁻¹ appeared in the sample with a C/H ratio of 0.3%, although its intensity was significantly lower than that observed at 0.9%, indicating the presence of a small amount of sp²-bonded carbon.

  1. The horizontal axis of the CV profile should indicate the standard potential to identify its position relative to the hydrogen reduction potential (H2/H+).

Our response:

Thank you for your comments. We’ve revised the horizontal axis label in Figure 6 as following reviewer’s comments.

  1. In addition, the author should indicate what reference potential +2.8V of forming .OH free radical is based on? If based on the reported CV profile, wouldn't the .OH free radical that can be generated occur in all C/H samples? (Every sample shows strong oxidation reaction beyond around 1.5 V, which is smaller than the 2.8 V mentioned in the manuscript, so every sample can generate .OH free radical at +2.8 V vs. the reported CV profile reference potential).

The oxidation position where .OH free radical occurs vs. the H2/H+ potential must be clearly defined. It is completely wrong to use the cell voltage of 2.88 to explain that it includes 2.8V. "Voltage" (Difference of potential) and "potential" (position) is totally a different thing. Anyway, the description here is not clear and is not supported by the shown data.

Our response:

Thank you for your insightful comment. To clarify, hydroxyl radicals (•OH), known as strong oxidants, have a high oxidation potential of +2.8 V. As reviewer pointed out, all sample has potential window value 2.8V, so this means all sample can generate OH free radical. In previous description, we considered the background current as well, which can harmful to stable generation of OH free radical. However, as commented by reviewer, we have revised sentence to avoid misunderstanding which explaining only 0.5 and 0.7% sample can generate OH free radical. Also, we have added a supporting reference to substantiate this point in the revised manuscript.

Additionally, although the CV analysis showed that all BDD samples exhibited relatively wide potential windows with no significant differences, the samples synthesized at C/H ratios of 0.3% and 0.9% exhibited higher background currents and unstable current behavior in the range of –1.0 to 1.0 V. This observation has been written in the revised manuscript to better reflect the electrochemical stability differences among the samples.

New reference: (Line 306)

  1. Najafinejad, M.S.; Chianese, S.; Fenti, A.; Iovino, P.; Musmarra, D. Application of Electrochemical Oxidation for Water and Wastewater Treatment: An Overview. Molecules 2023, 28.
  2. Qiao, J.; Xiong, Y. Electrochemical Oxidation Technology: A Review of Its Application in High-Efficiency Treatment of Wastewater Containing Persistent Organic Pollutants. Journal of Water Process Engineering 2021, 44.

Original: (Line 286)

According to the CV analysis, the BDD thin films deposited at C/H ratios of 0.5% and 0.7% exhibited the widest potential windows, indicating superior electrochemical stability. By contrast, at C/H ratios of 0.3% and 0.9%, the potential windows were comparatively narrower. The background current, reflected by the separation between the anodic and cathodic scans, increased, and the potential window was narrowed.

Revised: (Line 286)

According to the CV analysis, the BDD thin films deposited at C/H ratios of 0.5% and 0.7% exhibited the widest and stable potential windows, indicating superior electrochemical stability. By contrast, although the potential window values show no significant difference, C/H ratios of 0.3% and 0.9% samples exhibited higher background currents and unstable current behavior within the range of –1.0 to 1.0 V. This slightly affected the effective potential window and indicates a reduction in electrochemical stability.

Original: (Line 302)

Therefore, a C/H ratio of 0.7%, which exhibited the widest potential window (2.88 V), was confirmed to possess excellent electrochemical stability. In addition, because hydroxyl radicals (•OH), known as strong oxidants, are generated at approximately +2.8 V, a C/H ratio of 0.7% ensures a sufficiently wide anodic potential range for their formation. Accordingly, a C/H ratio of 0.7% is considered the optimal condition for BDD to achieve superior electrochemical performance as an electrode material.

Revised: (Line 302)

Therefore, a C/H ratio of 0.7%, which exhibited a wide and stable anodic potential window of 2.88 V vs. SCE with low background current, demonstrated excellent electrochemical stability. Such electrochemical characteristics enable the stable formation of hydroxyl radicals (•OH), which are strong oxidants with an oxidation potential of approximately +2.8 V, while has low background current and ensuring electrode stability. Accordingly, a C/H ratio of 0.7% is considered the optimal condition for achieving superior electrochemical performance and stability of BDD as an electrode material.

Reviewer 2 Report

Comments and Suggestions for Authors

This is an experimental article investigating properties of boron doped HFCVD diamond synthesized at different ration of C/H in feed gas mixture. The goal of the article is to find optimal regime for synthesis of diamond electrodes for electrochemical applications. The article contains a lot of experimental data, is generally well written, provides valuable information for optimizing of HFCVD synthesis, and after minor corrections definitely may be published in “Micromachines”.

L62-66: “Although diamonds have different power sources for the CVD process, the deposition process generally follows the same fundamental theory … the substrate is heated by radiative heating from the filament” – here it should be clarified that heating “from the filament” applies only to HFCVD, not to the other “different power sources”.

L 119: “TMB gas contains a carbon source” – “TMB gas mixture contains a carbon source”

Fig5: Axis labels on “Carrier concentration” axis are unclear: it is not trivial to get where the “zero” level actually is; and putting mark at negative concentration value (-1) looks strange. Also, it is clear that carrier concentration at 0.3% C/H is “very low”, but how low it is, what is the actual value?

Author Response

This is an experimental article investigating properties of boron doped HFCVD diamond synthesized at different ration of C/H in feed gas mixture. The goal of the article is to find optimal regime for synthesis of diamond electrodes for electrochemical applications. The article contains a lot of experimental data, is generally well written, provides valuable information for optimizing of HFCVD synthesis, and after minor corrections definitely may be published in “Micromachines”.:

: Thank you very much for your kind and encouraging comments. We sincerely appreciate your positive evaluation of the significance and clarity of our work. The manuscript has been carefully revised in response to your suggestions. All changes have been highlighted in yellow background in the revised manuscript, and we have provided detailed, point-by-point responses to each of your comments below.

  1. L62-66: “Although diamonds have different power sources for the CVD process, the deposition process generally follows the same fundamental theory … the substrate is heated by radiative heating from the filament” – here it should be clarified that heating “from the filament” applies only to HFCVD, not to the other “different power sources”.

Our response:

We appreciate the reviewer’s comment. We agree that the previously used phrase, “radiative heating from the filament,” could be misleading if interpreted as applying to all CVD methods. Since this study specifically focuses on hot-filament CVD (HFCVD), we have revised the sentence at L62–66 to clarify that the described deposition mechanism pertains only to HFCVD.

Original: (Line 62)

Although diamonds have different power sources for the CVD process, the deposition process generally follows the same fundamental theory of diamond synthesis:

Revised: (Line 62)

In the case of HF-CVD, the diamond film is deposited according to the following fundamental steps of diamond synthesis:

  1. L 119: “TMB gas contains a carbon source” – “TMB gas mixture contains a carbon source”

Our response:

We agree with the reviewer that the revised wording more accurately reflects the fact that TMB is introduced as part of a gas mixture, rather than as a pure gas. Accordingly, the sentence has been corrected in the manuscript.

Original: (Line 119)

TMB gas contains a carbon source (CH3); thus, the carbon contributions from both CH4 and TMB were considered when calculating the C/H ratio.

Revised: (Line 119)

TMB gas mixture contains a carbon source (CH3); thus, the carbon contributions from both CH4 and TMB were considered when calculating the C/H ratio.

  1. Fig5: Axis labels on “Carrier concentration” axis are unclear: it is not trivial to get where the “zero” level actually is; and putting mark at negative concentration value (-1) looks strange. Also, it is clear that carrier concentration at 0.3% C/H is “very low”, but how low it is, what is the actual value?

Our response:

We appreciate the reviewer’s valuable comment. We acknowledge that the “Carrier concentration” axis labeling in Figure 5 lacked clarity. In our study, we’ve doped boron as an impurity in diamond, which p-type impurity diamond. This means the carrier concentration should have positive value. Therefore, we’ve improved the axis labels by clearly indicating the zero level and removing inappropriate negative tick marks, as carrier concentration cannot be negative. In addition, the carrier concentration at a C/H ratio of 0.3% was measured to be approximately 0.05 × 1020 cm-3, and to improve clarity, the measured values are tabulated in Table 4.
